# Thermochemical Study of the Interaction of Cytosine and Uracil with Peptides in a Buffered Saline: Complex Formation with beta-Endorphin 30-31 (Human), L-Glutathion (Reduced) and α-L-Alanyl-L-Tyrosine

**DOI:** 10.3390/ijms24119764

**Published:** 2023-06-05

**Authors:** Vladimir P. Barannikov, Valeriy I. Smirnov, Igor N. Mezhevoi, Damir R. Koltyshev

**Affiliations:** G.A. Krestov Institute of Solution Chemistry, Russian Academy of Science, 153045 Ivanovo, Russia; vis@isc-ras.ru (V.I.S.); inm@isc-ras.ru (I.N.M.); kdr@isc-ras.ru (D.R.K.)

**Keywords:** uracil, cytosine, beta-endorphin 30-31, glutathion, alanyl tyrosine, molecular complexes, thermodynamics

## Abstract

The complex formation of uracil and cytosine with glycyl-L-glutamic acid (β-endorphin 30-31), γ-L-glutamyl-L-cysteinyl-glycine (glutathione reduced), α-L-alanyl-L-tyrosine, and α-L-alanyl-α-L-alanine in a buffered saline has been studied using dissolution calorimetry. The values of the reaction constant, the change in Gibbs energy, enthalpy, and entropy were obtained. It is shown that the ratio of the enthalpy and entropy factors depends on the charge of the peptide ion, and the number of H-bond acceptors in the peptide structure. The contributions of interaction between charged groups and polar fragments, hydrogen bonding, and stacking interaction are discussed, taking into account the effect of solvent reorganization around the reactant molecules.

## 1. Introduction

The interaction of proteins with various ligands, which include enzymes, hormones, and nucleic acids, contributes to the structural and regulatory functions of many biological objects. Unsurprisingly, a lot of biochemical and physicochemical research is aimed at a comprehensive study of these interactions. Recently, the complexation between proteins and DNA, which underlie the effect of viruses on a living organism, has attracted the attention of researchers [1,2,3,4,5,6,7]. Due to the compact structure of the viral particle, proteins and nucleic bases are spatially close, due to which a large number of bonds and interactions arise (hydrophobic, hydrogen bonding, salt bridges, etc.). Elucidating the fine molecular inhibitory mechanisms of these processes requires the study of factors that provide the stability of the formed complexes of proteins with nucleic acids. The complexity of the structure of macromolecules makes it difficult to solve this problem. Therefore, the study of model compounds of proteins and RNA has become widespread, which makes it possible to analyze the thermodynamic and structural aspects of their interactions under conditions of physiological pH values in aqueous solutions. It is relevant to study the mechanism and forces motivating the interaction between heterocyclic nucleic acid base molecules as RNA fragments and peptides as protein fragments. This work aims to study the influence of the structure and ionic state of peptides on the patterns of their interaction with nucleic acid bases in a buffered saline medium. To achieve this, the following was necessary:(a)To obtain thermochemical characteristics of the interaction of uracil and cytosine with a number of peptides of various structures by calorimetry of dissolution;(b)To calculate the values of the complexation constant, and the change in Gibbs energy, enthalpy, and entropy from calorimetric data;(c)To study the effect of the charge and the number of donors and acceptors of the H-bond in the structure of peptide on the patterns of binding to nucleic bases.

The objects of this study were uracil and cytosine as nucleic bases of the pyrimidine series, the structures of which are shown in Figure 1. In a neutral solution at pH < 9.5, uracil and cytosine take a lactam molecular form [8]. Peptides with different compositions and structures were chosen for the study. Among them, beta-endorphin 30-31 (human) is a C-terminal dipeptide of human beta-endorphin, whose chemical name is glycyl-L-glutamic acid (GlyGlu). Further, glutathione (reduced) is a tripeptide, the chain composition is described as gamma-L-glutamyl-L-cysteinyl-glycine (GluCysGly), and L-α-Alanyl-L-tyrosine is a dipeptide (AlaTyr) containing an aromatic phenolic group in the side chain. These peptides are involved in many metabolic processes, such as the biosynthesis, thiol protection, and redox regulation of cellular thiol proteins under oxidative stress, and conjugation to lipophilic xenobiotics. The peptides take a multicharged ionic form in an aqueous solution due to the processes of hydrolysis and acid dissociation. The structures of the ionic forms of the peptides that dominate in a neutral aqueous solution are shown in Figure 2. In the structure of these di- and tripeptides, oppositely charged terminal groups are distant from each other, unlike amino acids. Such a structure produces favorable steric conditions for the interaction of a bidentate peptide with polar nucleobase groups located on opposite sides of the aromatic ring. The selected peptides differ from each other in chain length, charge distribution, and the number of H-bond donors and acceptors in the molecule.

## 2. Results and Discussion

### 2.1. Distribution of Ionic Forms of Peptides

The mole fractions of ionic forms coexisting in an aqueous solution of the studied peptides at various pH values were determined based on the known acid dissociation constants by using the RRSU software (the creators are Vasiliev, V.P.; Borodin, V.A.; Kozlovsky, E.V.; Moscow, Russia) [9]. We used the following data: p*K*º_1_ = 2.91, p*K*º_2_ = 4.50, p*K*º_3_ = 8.21 for GlyGlu at *T* = 298.15 K [10]; p*K*º_1_ = 2.05, p*K*º_2_ = 3.49, p*K*º_3_ = 8.65, p*K*º_4_ = 9.60 for γ-GluCysGly at *T* = 298.15 K [11]. By extrapolating the results [12] for AlaTyr in a mixed (1,4-dioxane + water) solvent, the following values were obtained for an aqueous solution: p*K*º_1_ = 2.81; p*K*º_2_ = 7.92. The influence of the ionic strength of the solution on the dissociation constants of peptides is insignificant. As shown earlier for the GlyGlu peptide [10], the change in p*K* values for the stepwise dissociation is 0.2 in magnitude when the ionic strength varies from 0 to 1.0 mol dm^−3^. Therefore, we can assume that varying the ionic strength does not significantly affect the distribution of ionic forms of peptides in solution. In all experiments performed in this work, the ionic strength was constant, equal to the ionic strength of saline, 0.15 mol dm^−3^. The results of calculating the distribution of ionic forms in solution at various pH values are shown in Figure 3. As follows from the diagram, two anionic forms of the peptides dominate in a neutral solution at pH = 7.4: the mole fractions of ^+^GlyGlu^2−^ (marked as ^+^HL^2−^ in Figure 3) and GlyGlu^2−^ (L^2−^) anions are 0.86 and 0.13, and similar values for ^+/−^GluCysGly^−^ (^+^H_2_L^2−^) and ^−^GluCysGly^−^ (HL^2−^) are equal to 0.95 and 0.05, respectively. The ^+^AlaTyr^−^ (^+^HL^−^) zwitter ion is the dominant species (mole fraction is 0.77) in an aqueous solution of L-α-alanyl-L-tyrosine at pH = 7.4. The mole fraction of each ionic form of the peptides remained constant in a buffered saline medium with pH = 7.4 during the experiments. The chemical structures of the ^+^GlyGlu^2−^ and ^+/−^GluCysGly^−^ dominant anions and ^+^AlaTyr^−^ zwitter ion are presented in Figure 2.

### 2.2. Enthalpies of Dissolution of Peptides

All peptide dissolution enthalpies measured in this work are presented in Table 1. The enthalpies of dissolution in a phosphate-buffered saline differ from those for dissolution in pure water: Δ_sol_*H* = 10.84 kJ·mol^−1^ for GlyGlu; Δ_sol_*H* = 10.72 kJ·mol^−1^ for AlaTyr (measured in this work); Δ_sol_*H* = 20.85 kJ·mol^−1^ for γGluCysGly [13]. The dissolution of GlyGlu and AlaTyr in buffered saline is accompanied by a greater endothermic effect than the process in water. The dissolution effect of the γGluCysGly tripeptide in buffered saline is less endothermic than in water. The observed differences may be associated with a change in the ionic state of the peptides and the contribution of their interaction with electrolytes in a buffered saline.

The enthalpies of dissolution of peptides noticeably change when nucleic bases are added to the buffer solution. Changes in enthalpy depend on the concentration of the nucleic base. The difference between the enthalpies of dissolution in a buffered saline solution Δ_sol_*H^m^*_(buffer)_ and in the same solution with additions of a nucleic base Δ_sol_*H^m^*_(buffer+NB)_ can be considered as a thermochemical characteristic of the interaction between a peptide and a nucleic base Δ_tr_*H^m^*.
Δ_tr_*H^m^* = Δ_sol_*H^m^*_(buffer+NB)_ − Δ_sol_*H^m^*_(buffer)_(1)

The dependences of Δ_tr_*H^m^* values on the molar ratio of the nucleic base and peptide at a constant *m*_pept_ = 0.003 mol·kg^−1^ are shown in Figure 4. As can be seen, the interaction of peptides with uracil and cytosine is usually accompanied by an exothermic effect, with the exception of the interaction of AlaTyr with cytosine, for which an endothermic effect is observed. The shape of the curves in Figure 4 demonstrates the nonlinear dependence of Δ_tr_*H^m^* on the concentration of the nucleic base. The Δ_tr_*H^m^* values change most sharply with small additions of a nucleic base, and the changes are fading when *m*_NB_/*m*_pept_ > 4. In cases of interaction of peptides with uracil and interaction of γGluCysGly with cytosine, the Δ_tr_*H^m^* values change until almost constant values are reached. Such behavior suggests complex formation in the studied systems. Now there is information about the structure of a number of protein-DNA and protein-RNA complexes [14]. The binding of proteins to DNA was established based on the results of UV spectroscopy, mass spectrometry, and phase transition temperatures. However, the mechanism and driving forces of complex formation have not been sufficiently disclosed even for the interaction between the structural units of these biomacromolecules, i.e., between amino acids (or short peptides) and nucleobases. The formation of complexes between the protein model and nucleic acid bases has been theoretically proven by the B3LYP method [15,16], and a significant contribution of hydrogen bonds and stacking interactions to the binding of amino acids to nucleobases has been established [16,17]. It is known that the NH fragment at the 3rd position of the uracil cycle and the NH_2_ group at the 4th position of cytosine cycle exhibit the ability of an H-bond donor. The O-4 atom of uracil and N-3 and O-2 atoms of cytosine can act as H-bond acceptors.

### 2.3. Thermodynamic Parameters of Complexation of Peptides with Nucleic Bases

The Δ_tr_*H^m^* values formed a basis for calculating the thermodynamic functions of complex formation. Apparent complexation constants and enthalpy changes (lg*K_r_* and Δ_r_*H*) were calculated using the initial total concentrations of reagents and experimental values of Δ_tr_*H^m^* by means of the computer program HEAT [18,19]. For peptides, total concentrations were used, which include all ionic forms coexisting in solution. The search for unknown parameters (lg*K*_r_, ∆_r_*H*) is reduced to the minimization of the F-functional given by
(2)$$F=\sum_{i=1}^{N}w_{i}{\left(\Delta_{\mathrm{tr}}H_{i}^{m}{}_{\exp}-\Delta_{\mathrm{tr}}H_{i}^{m}{}_{\mathrm{calc}}\right)}^{2}$$
where Δ_tr_*H_i_* is the enthalpy effect from the *i*-th reaction, *N* is the number of experiments, and *w_i_* is a weighted factor. Thus, mathematical treatment of the Δ_tr_*H^m^* = f(*m*_NB_) dependences using the minimization computer program HEAT allows the enthalpy and apparent constant of complex formation to be simultaneously found. The binding stoichiometry was also a fitting parameter when processing the results. The best agreement between the experimental and calculated Δ_tr_*H^m^_i_* values was achieved at 1:1 stoichiometry of nucleic base–peptide complexes. All the calculated thermodynamic parameters of complex formation are summarized in Table 2.

The constants of complexation of the studied peptides with uracil and cytosine in buffered saline are small; the values of lg*K_r_* take values in the range of 0.95 to 2.0 in magnitude, which is typical for many reversible biochemical processes. The ability of nucleic bases to reverse complexation reactions is a necessary and important condition to perform their biological functions. Complexes of uracil exhibit a greater stability compared to cytosine complexes. Among the uracil complexes, the smallest constant appears for the interaction with the aliphatic peptide AlaAla. As can be seen from Table 1, the enthalpies of dissolution of the peptide change insignificantly with varying concentration, which leads to the lowest positive values of the enthalpies of transfer. Small values of Δ_tr_*H^m^* make it difficult to study the complexation with this peptide, and reliable data could not be obtained in the case of cytosine.

The effect of the structure of the peptide and the nucleobase on the thermodynamic parameters of complexation is manifested in the ratio of the enthalpy and entropy factors of the process. A correct interpretation of the patterns of complex formation in solution is possible only if the effect of solvent reorganization around the reactant molecules is taken into account. For this purpose, the model of overlapping hydrate co-spheres [20,21] has proved to be suitable. The penetration of charged groups of peptides into the hydration shells of nucleic bases can lead to partial dehydration of the NH group and two O-atoms in the case of uracil, or O, N-atoms and NH_2_-group in the case of cytosine. The NH-fragment in the uracil cycle forms a more stable hydration shell structured by H-bonds with water molecules, unlike the shell around the N-heteroatom in the cytosine cycle. Thus, the set of reaction centers capable of forming complexes with peptides means we can expect that their dehydration will be accompanied by a greater expenditure of energy and a greater increase in disorder in the case of uracil compared to cytosine. From the data in Table 3, it follows that the complexation of ^+^GlyGlu^2−^ and ^+/−^GluCysGly^−^ anions, and ^+^AlaAla^−^ zwitter-ions with uracil is controlled by the entropy factor, −Δ_r_*H* < *T*Δ_r_*S*. During complexation, water molecules move from the hydration shells of the polar groups of uracil into the solvent bulk. A significant increase in entropy during the formation of complexes proves the predominance of the contribution of the dehydration of polar groups in this case. The formation of complexes with cytosine is controlled by the enthalpy factor, −Δ_r_*H* > *T*Δ_r_*S*, which indicates the predominance of the exothermic contribution from the interactions between polar groups and the formation of H-bonds between the peptide and cytosine.

In the case of the ^+^AlaTyr^−^ zwitter-ion, the side chain of which is an aromatic ring with a hydroxo group, the complexation exhibits inverse ratios of Δ_r_*H* and *T*Δ_r_*S* factors. This may be connected with the fact that tyrosine residue is capable of stacking interaction with the nucleic base ring. Other peptides of the studied series do not possess such an ability. The complexation with ^+^AlaTyr^−^ is an exothermic process when interacting with uracil and, conversely, an endothermic process when interacting with cytosine. The endothermic effect in the latter case can be attributed to the contribution of stacking interactions of AlaTyr with Cyt in an aqueous solution, accompanied by a dominant endothermic contribution of the partial dehydration of aromatic rings of the reactants. The greatest favorable entropy effect *T*Δ_r_*S* = 17.15 kJ·mol^−1^ confirms this assumption. The interaction of AlaTyr with uracil, in contrast, proceeds due to a favorable enthalpy factor, −Δ_r_*H* > *T*Δ_r_*S*. It is likely that the stacking interaction does not significantly contribute to the complexation with uracil.

A comparison of the results for the studied complexes shows the effect of the charge of peptide ions on the enthalpies of binding to neutral nucleic bases. The interaction with tripolar anions (^+^GlyGlu^2−^, ^+/−^GluCysGly^−^) having two charged carboxylate groups in the structure is an exothermic process. Moreover, the exothermic effects of interaction with cytosine are greater than the effects characteristic of uracil. The greatest exothermic effect, Δ_r_*H* = −12.7 kJ·mol^−1^, manifested during the interaction of Cyt with ^+^GlyGlu^2−^, is accompanied by the most unfavorable entropy effect, *T*Δ_r_*S* = −7.3 kJ·mol^−1^, and this results in the minimum value of the reaction constant (lg*K_r_* = 0.95).

The influence of the number of H-bond acceptors in the peptide structure can be noted. Thus, complexation with the ^+/−^GluCysGly^−^ anion having six acceptor centers (four O-atoms in the carboxylate groups and two O-atoms in the amide fragments) proceeds with the highest reaction constant (lg*K_r_* values are equal to 2.01 and 1.39) compared to other peptides having from three to five centers.

Summing up the results obtained, we can highlight the trend of compensation of enthalpy and entropy factors in the process of complexation of peptides with nucleic bases, which is described by the following relation:*T* Δ_r_*S* = (7.0 ± 0.5) + (1.0 ± 0.07) × Δ_r_ *H*,
where the standard deviation is 1.2 kJ·mol^−1^ and the correlation coefficient is 0.99016 at a confidence level of 0.95.

## 3. Materials and Methods

The list of chemicals used is presented in Table 3. The chemicals were dried under a vacuum at *T* = 343.15 K for 24 h. The water content was determined using the Karl Fischer method (Volumetric KF Titrator V30, Mettler Toledo) and amounted to 0.001 mass fraction for Ura and Cyt, and for GlyGlu, GluCysGly, and GlyTyr, the water content was 0.003, 0.002, and 0.002, respectively. All measurements were performed in an aqueous phosphate-buffered saline. A buffered saline, pH = 7.4, was prepared by dissolving five tablets (Sigma, USA) in 1 dm^3^ of purified water at *T* = 298 K. The solution contained 0.0073 mol dm^−3^ Na_2_HPO_4_; 0.0046 mol dm^−3^ KH_2_PO_4_; 0.137 mol dm^−3^ NaCl and 0.0027 mol dm^−3^ KCl. The ionic strength of the solution was 0.15 mol dm^−3^. The peptide and nucleic base samples were weighed with an uncertainty of 1 × 10^−5^ g. The pH index of the solutions was checked using a digital Five-Easy pH-meter (Mettler Toledo), with a standard uncertainty of 0.02 at T = 298.15 K. The buffered saline was used as solvent for the preparation of sample solutions by mass (with an accuracy of ±1 × 10^−5^ g) using a Sartorius-ME21 analytical balance. The peptide concentration remained constant at the level *m*_pept_ = 0.003 mol∙kg^−1^. The molalities of nucleic bases varied within the range of 0.0015 to 0.06 mol∙kg^−1^. The standard uncertainty in the molality of the solutions was estimated within ±2 × 10^−4^ mol·kg^−1^.

Calorimetric measurements were made with an ampoule-type “isoperibol” calorimeter with a 60 cm^3^ reaction vessel and electrical calibration at *T* = 298.15 ± 0.01 K and *P* = 100.5 ± 0.7 kPa, which has been described in detail previously [22]. Electrical calibration of the calorimeter was performed after each experiment. The heat effect from sample dissolution was compared with the effect from the calibrated Joule heating using the digital Standard Temperature Measuring Instrument (BIC, Minsk, Belarus). The calorimetric device’s accuracy was checked prior to the experiments by measuring the enthalpy of the KCl dissolution in water. The experimentally observed value Δ_sol_*H*° = (17.52 ± 0.06) kJ·mol^−1^ agrees well with the ICTAC working group suggestion (17.47 ± 0.07) kJ·mol^−1^ [23]. In each experiment, a constant amount of the peptide was dissolved in the buffer solution of a nucleic base with a concentration varying from 0.003 to 0.06 mol kg^−1^. The heat effects of dissolution ranged from 1.5 to 3.5 J. The relative error in the dissolution enthalpy measurements did not exceed 0.5%. Each experiment was repeated at least two times, achieving reproducibility of the enthalpy of dissolution within the standard uncertainty of u(Δ_sol_*H^m^*) = ±0.005 × Δ_sol_*H^m^*.

## 4. Conclusions

The resulting set of thermodynamic characteristics of the complexation of nucleic bases (uracil and cytosine) with a number of peptides allows us to note some important patterns of this process. Complexes of uracil and cytosine with peptides show low stability in a buffered saline medium, which is typical for many reversible biochemical reactions. The values of the logarithm of the complexation constant vary from 0.9 to 2.0 in magnitude.

The effect of the structure of the peptide and the nucleobase on the thermodynamic parameters of complexation is manifested in the ratio of the enthalpy and entropy factors. The complexation with uracil in most cases is controlled by the entropy factor, −Δ_r_*H* < *T*Δ_r_*S*, and the enthalpy factor dominates when interacting with cytosine, −Δ_r_*H* > *T*Δ_r_*S*. The inverse ratios of the factors Δ_r_H and TΔ_r_S are observed in the case of ^+^AlaTyr^−^ zwitter ions, which are capable of stacking interaction with nucleic bases.

The charge of the peptide ion affects the enthalpies of binding with neutral nucleic bases. The complexation with tripolar anions of peptides having two charged carboxylate groups in the structure is accompanied by a greater exothermic effect than the interaction with zwitterions. This trend proves that contact between charged groups of peptides and polar groups of nucleic bases in solution produce an exothermic contribution to the reaction enthalpy.

The dependence of the complexation constants on the number of H-bond acceptors in the peptide structure shows a significant contribution of H-bonding to the stabilization of complexes of peptides with nucleic bases. The highest reaction constants were found for the tripeptide with six acceptor centers in the structure.

## Figures and Tables

**Figure 1 ijms-24-09764-f001:**
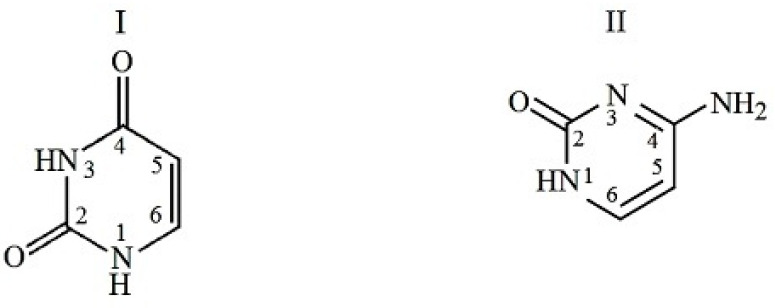
Chemical structure of uracyl (I) and cytosine (II).

**Figure 2 ijms-24-09764-f002:**
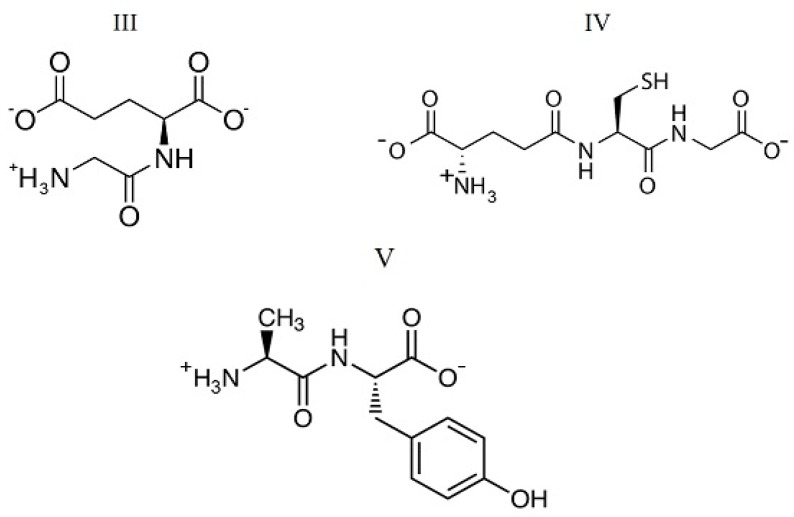
Dominant anionic forms of glycyl-L-glutamic acid (III) and L-γ-glutamyl-L-cysteinyl-glycine (IV) and zwitter ion of L-alanyl-L-tyrosine (V).

**Figure 3 ijms-24-09764-f003:**
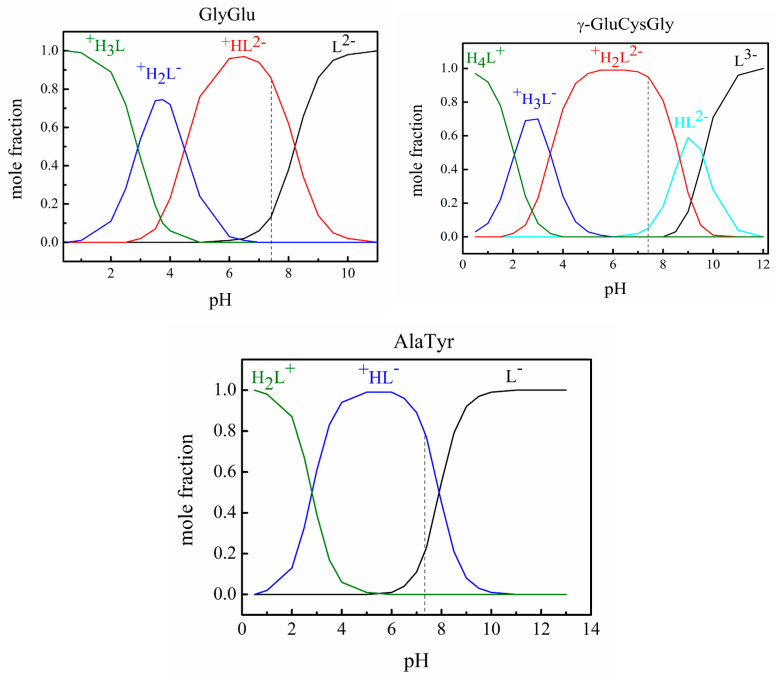
**A** diagram of the distribution of ionic forms of glycyl-L-glutamic acid, L-γ-glutamyl-L-cysteinyl-glycine, and alanyl-L-tyrosine depending on pH at *T* = 298.15 K; the dotted line marks the value of pH = 7.4.

**Figure 4 ijms-24-09764-f004:**
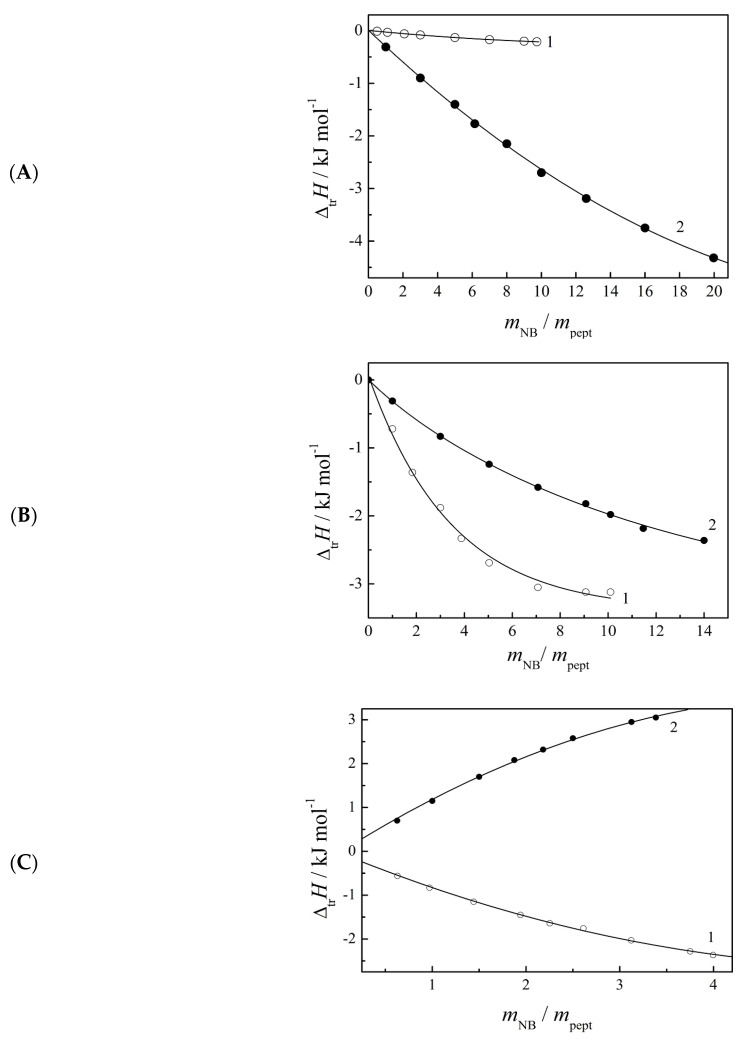
The transfer enthalpies of peptides ((**A**)—GlyGlu; (**B**)—γGluCysGly; (**C**)—AlaTyr) from buffered saline to buffer solutions of (1) uracil and (2) cytosine at different ratios of nucleic base and peptide, *m*_pept_ = 0.003 mol·kg^−1^, *T* = 298.15 K, pH = 7.4.

**Table 1 ijms-24-09764-t001:** The enthalpies of peptide dissolution at constant peptide molality (*m* = 0.003 mol·kg^−1^) in phosphate-buffered saline with additives of uracil and cytosine (Δ_sol_*H^m^*, kJ·mol^−1^) at *T* = 298.15 K; *p* = 100.5 kPa; pH = 7.4; and molality of nucleic bases *m*_NB_, mol·kg^−1^.

*m* _NB_	Δ_sol_*H^m^*	*m* _NB_	Δ_sol_*H^m^*
GlyGlu + uracil		GlyGlu + cytosine	
0	11.43	0.0030	11.12
0.0015	11.42	0.0091	10.53
0.0033	11.39	0.0153	10.03
0.0062	11.37	0.0185	9.66
0.0090	11.35	0.0242	9.28
0.0151	11.30	0.0303	8.73
0.0212	11.25	0.0378	8.24
0.0272	11.20	0.0480	7.68
0.0292	11.21	0.0599	7.11
γGluCysGly + uracil		γGluCysGly + cytosine	
0	20.36	0.0030	20.05
0.0030	19.64	0.0090	19.53
0.0055	19.00	0.0150	19.12
0.0090	18.48	0.0210	18.78
0.0116	18.03	0.0240	18.64
0.0151	17.67	0.0300	18.38
0.0212	17.24	0.0344	18.18
0.0272	17.21	0.0420	17.98
0.0303	17.28		
AlaTyr + uracil		AlaTyr + cytosine	
0	10.89	0.0050	11.59
0.0050	10.33	0.0080	12.04
0.0078	10.06	0.0115	12.59
0.0115	9.74	0.0155	12.97
0.0155	9.44	0.0175	13.21
0.0180	9.27	0.0201	13.47
0.0201	9.13	0.0250	13.84
0.0250	8.86	0.0271	13.94
0.0300	8.61		
0.0320	8.53		
AlaAla + uracil			
0	−7.46		
0.0050	−7.44		
0.0078	−7.41		
0.0115	−7.36		
0.0155	−7.32		
0.0201	−7.27		
0.0250	−7.25		
0.0300	−7.24		
0.0320	−7.23		

Note. Standard deviation (*u*) of experimental parameters: *u*(*m*) = 0.00005 mol·kg^−1^; temperatures u(*T*) = 0.01 K and *u*(*p*) = 0.7 kPa; u(Δ_sol_*H^m^*) = ± 0.005 × Δ_sol_*H^m^*. The molalities of peptides and nucleic bases are calculated per kilogram of buffer saline.

**Table 2 ijms-24-09764-t002:** Thermodynamic parameters of complexation of peptides with uracil and cytosine in buffered saline at pH = 7.4 and *T* = 298.15 K.

	lg*K_r_* *	Δ_r_*G*	Δ_r_*H*	*T* Δ_r_*S*
		kJ·mol^−1^		
Ur + GlyGlu	1.16 ± 0.001	−6.64 ± 0.01	−0.7 ± 0.1	5.9 ± 0.1
Ur + γ-GluCysGly	2.01 ± 0.04	−11.4 ± 0.2	−4.4 ± 0.1	7.0 ± 0.3
Ur + AlaTyr	1.45 ± 0.001	−8.3 ± 0.01	−5.30 ± 0.05	3.0 ± 0.06
Ur + AlaAla	1.03 ± 0.05	−5.9 ± 0.3	1.0 ± 0.04	6.9 ± 0.3
Cyt + GlyGlu	0.95 ± 0.001	−5.40 ± 0.01	−12.7 ± 0.08	−7.3 ± 0.1
Cyt + γ-GluCysGly	1.39 ± 0.002	−7.89 ± 0.01	−4.79 ± 0.07	3.1 ± 0.1
Cyt + AlaTyr	1.29 ± 0.002	−7.35 ± 0.01	9.8 ± 0.1	17.15 ± 0.1

* unit of *K_r_* is kg·mol^−1^.

**Table 3 ijms-24-09764-t003:** The list of chemicals, their origin, and purity values.

Chemical ^a^	M ^b^	CAS No. ^c^	Origin	Purity ^d^
Uracil	112.09	66-22-8	Termo Ficher Scientific	0.996
Cytosine	111.10	70-30-7	Appolo Scientific	0.996
Glycyl-L-glutamic acid (beta-endorphin 30-31 human)	204.18	7412-78-4	Bachem	>0.99
L-γ-Glutamyl-L-cysteinyl-glycine (glutathion reduced)	307.32	108457-42-7	Tokyo Chemical Industry	>0.99
L-α-Alanyl-L-tyrosine	252.27	3061-88-9	Abcr GmbH	>0.98

^a^ All reagents were dried in vacuum; ^b^ Molar mass of chemical, kg·kmol^−1^; ^c^ Chemical Abstract Service registry number; ^d^ Mass fraction (as stated by the supplier).

## Data Availability

All new data obtained in this work are contained in the tables presented in the article.

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
