# Peer review of "Thermochemical Study of the Interaction of Cytosine and Uracil with Peptides in a Buffered Saline: Complex Formation with beta-Endorphin 30-31 (Human), L-Glutathion (Reduced) and α-L-Alanyl-L-Tyrosine"

_ijms, 2023, doi:10.3390/ijms24119764_

Round 1
Reviewer 1 Report
The article is devoted to investigation of complexation between nucleobases and oligopeptides served as model compounds. Thermodynamic values of 1:1 complex formation were derived from dissolution calorimetry data. The influence of dipeptide charge, number of acceptor centers and possibility of stacking interactions was discussed during interpretation of binding constants, reaction enthalpies and entropies. The article provides valuable information about interaction between model compounds that can be used in further understanding of more complex systems. The manuscript is well-organized. However, some minor changes are able to improve it for a reader.
Table 1: Please, add pH value to table description.
Line 151: space after «Table» is omitted
Line 175-176, 193-194: there is a strange space between the lines
Line 261: r should be in subscript in ΔrH and ΔrS
The possible influence of ionic strength to obtained results, in particular, distribution diagram, should be discussed.
In tables 1 and 2 complexation between Ur and AlaAla is included. However, AlaAla dipeptide absent in Fig. 3 and respective discussion (lines 73-88). It should be mentioned as minimum in the text. Why the 8th possible combination between Cyt and AlaAla was not studied or discussed?
For both GlyGlu and AlaTyr two different forms exist at pH = 7.4. As I understand, only one complexation process was treated in each case. I recommend include discussion of the possible effect of complexation between nucleobases and different forms of dipeptides on calculated thermodynamic parameters and their interpretation.
Author Response
Table 1: Please, add pH value to table description.
1. The pH value was added to the heading of Table 1.
2. The omitted space restored on line 151.
3. Extra spaces in lines 175-176, 193-194 are removed.
4. A subscript has been used for the "r" symbol in line 261.
5. The discussion in section 2.1 has been expanded as follows.
“The influence of the ionic strength of the solution on the dissociation constants of peptides is insignificant. As shown earlier for the GlyGlu peptide, the change in pK values for the stepwise dissociation is 0.2 in magnitude when the ionic strength varies from 0 to 1.0 mol dm-3. Therefore, we can assume that varying the ionic strength does not significantly affect the distribution of ionic forms of peptides in solution. In all experiments performed in this work, the ionic strength was constant, equal to the ionic strength of saline, 0.15 mol dm-3.”
6. Section 2.3 has been supplemented with the following comment.
“Among the uracil complexes, the smallest constant appears for the interaction with the aliphatic peptide AlaAla. As can be seen from Table 1, the enthalpies of dissolution of the peptide change insignificantly with varying concentration, which leads to the lowest positive values of the enthalpies of transfer. Small values of ΔtrHm make it difficult to study the complexation with this peptide, and reliable data could not be obtained in the case of cytosine.”
7. The resulting complexation constants are apparent, which are obtained using the total concentrations of peptides in solution. All ionic forms of the peptide participate in the formation of complexes, but, of course, the dominant form in the solution makes the greatest contribution.
The term has been defined in Section 2.3.
“Apparent complexation constants and enthalpy changes (lgKr and ΔrH) were calculated using the initial total concentrations of reagents and experimental values of ΔtrHm by means of the computer program HEAT [18,19]. For peptides, total concentrations were used, which include all ionic forms coexisting in solution.”
Reviewer 2 Report
The manuscript "Thermochemical Study of the Interaction of Cytosine and Uracil with Peptides in a Buffered Saline: Complex Formation with beta-Endorphin 30-31 (Human), L-Glutathion (Reduced) and α-L-Alanyl-L-Tyrosine" describes the results of thermochemical experiments consisting in dissolution of different peptides in buffer solutions of two nitrogen bases of DNA and RNA. Basing on the primary experimental data, the thermodynamic characteristics of the reactions between peptides and nitrogen bases were calculated including changes in Gibbs energy, enthalpy and entropy. The interpretation of the obtained data was given basing on the literature results of quantum chemical calculations and other reasoning. In general, the paper can be recommended for publication as it provides basis for further investigation of the interactions between nucleic acids and proteins, which result in, for example, such important structures as chromatin and telomeres. However, there are some minor comments:
1. Reference 9 is irrelevant as it leads to "Page not found". It is better to use other reference (or other analogous software).
2. How the corrected temperature rise (decrease) in isoperibol calorimeter was calculated? Was a Regnault–Pfaundler method used or some other technique?
3. How the correction on infinite dilution of was implemented? It seems that the infinite dilution can potentially be a reason behind the total change in enthalpy in some experiments (e.g. U + GlyGly or AlaAla where the total delta H is ~0.2 kJ mol-1).
4. How many times was every data point repeated? The type on uncertainty specified does not imply that the experiments were even duplicated.
Author Response
- The slash “ / ” at the end of the specified erroneous address (https://chemequ.ru/online-progs/rrsu-online) was missing in reference [9]. Link https://chemequ.ru/online-progs/rrsu-online/ is available. Reference 9 has been corrected as follows.
- Vasiliev, V.P., Borodin, V.A., Kozlovsky E.V. The use of computers in chemical-analytical calculations. Moscow, Higher School, 1993, 110 pp. (Russia).
RRSU-online https://chemequ.ru/online-progs/rrsu-online/.
2. Temperature changes in the calorimetric cell were measured by the comparative method using the digital Standard Temperature Measuring Instrument (BIC, Minsk). To do this, after each experiment, the calorimeter was calibrated by electric current. The heat effect from sample dissolution was compared with the effect from the calibrated Joule heating.
The following has been added to the manuscript.
“Electrical calibration of the calorimeter was performed after each experiment. The heat effect from sample dissolution was compared with the effect from the calibrated Joule heating using the digital Standard Temperature Measuring Instrument (BIC, Minsk).”
3. You are probably referring to nucleobase dilution, which is characterized by enthalpies of transfer, and the comparison of enthalpies of transfer and their uncertainties. As indicated in the note under Table 1, the uncertainty in the enthalpy of dissolution is u(ΔsolHm) = ± 0.005×ΔsolHm, i.e. 0.5%. The enthalpies of transfer were calculated as the difference between the enthalpies of dissolution of the peptide in the buffer and in the buffer with the addition of the nucleobase.
ΔtrHm = ΔsolHm(buffer+NB) - ΔsolHm(buffer)
In this case, the largest absolute values of ΔtrHm and their uncertainties were:
GlyGlu+Ur: 11.21 - 11.43 = -0.22 ± 0.06 kJ/mol
GlyGlu + Cyt: 7.11 - 11.12 = 4.01 ± 0.06
γGluCysGly + Ur: 17.28 - 20.36 = -3.08 ± 0.1
γGluCysGly + Cyt: 17.98 - 20.05 = -2.07 ± 0.1
AlaTyr + Ur: 8.53 - 10.89 = -2.36 ± 0.05
AlaTyr + Cyt: 13.94 - 11.59 = 2.35 ± 0.07
AlaAla + Ur: -7.23 - (-7.46) = 0.23 ± 0.04 kJ/mol.
4. Yes, the uncertainty of the enthalpies of dissolution indicated in the manuscript is not determined from the statistics of experiments repeated on several occasions.
The enthalpies of dissolution were calculated from the calorimetric data using the equation
ΔsolHm = Q*M/g
where Q is the heat effect of dissolution in J, g and M are the mass and molar mass of the peptide. The relative uncertainty of the enthalpy of dissolution can be estimated as
δ(ΔsolHm) = δQ/Q + δg/g.
Dissolution heat effects ranged from 2 to 7 J with an uncertainty of 0.01 to 0.025 J. The mass of the solute was from 0.06 to 0.1 g with an uncertainty of 10-5 g. With these parameters, the standard relative uncertainty assigned to the values obtained for the molar enthalpy of dissolution did not exceed 0.005, and u(ΔsolHm) = ± 0.005×ΔsolHm.
Each experiment was repeated at least two times, achieving reproducibility of the enthalpy of dissolution within the declared uncertainty.
The following has been added to the manuscript.
“Each experiment was repeated at least two times, achieving reproducibility of the enthalpy of dissolution within the standard uncertainty of u(ΔsolHm) = ± 0.005×ΔsolHm.”